# EM-GANSim: Real-time and Accurate EM Simulation Using Conditional GANs for 3D Indoor Scenes

## Abstract

We present a novel machine-learning (ML) approach (EM-GANSim) for real-time electromagnetic (EM) propagation that is used for wireless communication simulation in 3D indoor environments. Our approach uses a modified conditional Generative Adversarial Network (GAN) that incorporates encoded geometry and transmitter location while adhering to the electromagnetic propagation theory. The overall physically-inspired learning is able to predict the power distribution in 3D scenes, which is represented using heatmaps. Our overall accuracy is comparable to ray tracing-based EM simulation, as evidenced by lower mean squared error values. Furthermore, our GAN-based method drastically reduces the computation time, achieving a 5X speedup on complex benchmarks. In practice, it can compute the signal strength in a few milliseconds on any location in 3D indoor environments. We also present a large dataset of 3D models and EM ray tracing-simulated heatmaps. To the best of our knowledge, EM-GANSim is the first real-time algorithm for EM simulation in complex 3D indoor environments. We plan to release the code and the dataset.

## 1 Introduction

Electromagnetic (EM) waves, characterized by the oscillation of electric and magnetic fields, are central to technologies such as visible light, microwave ovens, and wireless communication systems, including Wi-Fi and 5G. Maxwell's equations (Maxwell, 1873) describe the interaction and propagation of these electric and magnetic fields through space, forming the theoretical foundation for understanding EM wave behavior, including reflection, refraction, diffraction, and scattering. These principles are critical when modeling wave propagation in complex environments (Obaidat & Green, 2003; D'Aucelli et al., 2018), such as indoor spaces, where multiple interactions with obstacles and materials occur.

In the field of EM simulation, various methods are employed to understand wave propagation and interaction with media. Path loss and attenuation play crucial roles in these simulations, measuring how much signal power diminishes over distance, due to obstacles or the medium itself. Ray tracing is a widely-used technique for simulating wave interactions with surfaces (Bertoni et al., 1994; Seidel & Rappaport, 1992), as it balances computational efficiency and accuracy. It simulates rays, as narrow beams of EM energy, traveling in straight lines and accounting for key phenomena like reflection and diffraction. The method's efficiency makes it popular for 5G network planning (Hsiao et al., 2017), vehicular communications (Wang & Manocha, 2022), electromagnetic characterization (Egea-Lopez et al., 2021), and ground-penetrating radar (Zhang et al., 2006). Other methods such as wave-based methods that numerically solve Maxwell's equations can provide more accurate results, capturing complex wave behavior such as diffraction and scattering more precisely. However, these methods are often too computationally intensive for real-time or large-scale applications (Coifman et al., 1993; Taflove et al., 2005),

Despite its advantages, current EM simulation systems, particularly those based on ray tracing, have limitations in terms of handling dynamic scenes or complex environments. Ray tracing relies on modeling rays, i.e., narrow beams of EM energy, that travel in straight lines until they encounter an object, tracing their paths from a source and modeling interactions like reflections and diffrac-

tions (McKown & Hamilton, 1991). The simulation accuracy depends on detailed environmental models and material properties, making it computationally intensive, and needs significant processing power to simulate the numerous potential ray paths in complex environments. For dynamic scenes and detailed indoor environments, the need to continually update the models and recompute new paths in real-time is a major challenge. Indoor simulations are particularly difficult due to the complexity and density of the obstacles, which further increases the computational load, making current methods inefficient for applications requiring quick responses, such as 5G network planning, where higher frequencies and more complex environments are used (Rappaport et al., 2013; Wang et al., 2020).

Innovative solutions, such as integrating generative adversarial networks (GANs) into EM simulations, are being explored to address these limitations. GAN models include considerations for path loss, reflection, and diffraction in their loss functions, and they also account for material properties and multipath propagation, thereby improving simulation accuracy and heatmap generation for real-world applications.

**Main Results:** We present a novel GAN-based prediction scheme for real-time EM simulation in 3D indoor scenes. Our formulation uses a physically-inspired generator to predict wireless signal received power heatmaps and ensures high accuracy by incorporating detailed signal propagation mechanisms such as direct propagation, reflection, and diffraction. These physical constraints are embedded within the GAN's loss function to ensure that the generated data adheres to the principles of electromagnetic wave propagation. We use ray tracing techniques to model how signals propagate through an environment, considering reflections off surfaces and diffractions around the obstacles (Sangkusolwong & Apavatjrut, 2017). We evaluate these physical interactions using EM propagation models and the uniform theory of diffraction (UTD) (Kanatas et al., 1997) to predict the path loss for indoor environments accurately. Our approach not only improves the reliability of the heatmap predictions but also enhances the robustness and convergence of the GAN during training. Our main contributions include:

- *Accurate Power Distributions*: By employing conditional Generative Adversarial Networks (cGANs) and utilizing the strengths of physics-inspired learning, our approach can predict accurate power distributions in 3D indoor environments.

- *Real-Time Performance*: We highlight the performance on 15 complex 3D indoor benchmarks. Our approach significantly reduces the computational time needed for simulations compared to prior methods based on ray tracing. Our GAN models streamline the simulation process, achieving 5X faster running time on entire power map generation for various-sized indoor models. Additionally, it enables real-time simulation for individual data points in just a few milliseconds.

- *Dataset*: We present a large, comprehensive dataset featuring varied indoor scenarios (2K+ models) and simulated heatmaps (more than 64M) to train our model. This dataset ensures robust and generalized model performance across diverse conditions and is used for training and testing.

## 2 PRIOR WORK

Recent efforts in integrating ML with EM ray tracing and wireless communication systems have highlighted the potential of ML in enhancing wireless communication technologies in various ways. DeepRay (Bakirtzis et al., 2022) uses a data-driven approach that integrates a ray-tracing simulator with deep learning models, specifically convolutional encoder-decoders such as U-Net and SDU-Net, enhancing indoor radio propagation modeling for accurate signal strength prediction in various indoor environments. The model is able to learn from multiple environments and predict unknown geometries with high accuracy. WAIR-D (Huangfu et al., 2022) introduces a comprehensive dataset supporting AI-based wireless research, emphasizing the creation of realistic simulation environments for enhanced model generalization and facilitating fine-tuning for specific scenarios using real-world map data. Huang et al. (Huang et al., 2021) integrate ray tracing and an autoencoding-translation neural network to perform 3-D sound-speed inversion, improving efficiency and accuracy in underwater acoustic applications. Yin et al. (Yin et al., 2022) investigate the use of millimeter wave (mmWave) wireless signals in assisting robot navigation and employ a learning-based classifier for link state classification to enhance robotic movement and decision-making in complex

environments. There are other methods that combine deep reinforcement learning with enhanced ray tracing for antenna tilt optimization and those leveraging 5G MIMO data for beam selection using deep learning techniques to improve cellular network performance through efficient geospatial data processing and precise signal optimization (Zhu et al., 2022; Wang et al., 2023).

ML techniques have also been used to predict the received power in complex indoor and urban environments (Yun & Iskander, 2015). Traditional methods like regression models, decision trees, and support vector machines have been used to model the propagation characteristics of electromagnetic fields. The performance of these methods has been improved by adapting to data from specific environments, thereby enhancing prediction accuracy for both line-of-sight (LoS) and non-line-of-sight (NLoS) conditions (Filosa et al., 2016; Dong et al., 2020; Williams et al., 2015).

Despite their advancements, traditional simulation methods (e.g., ray tracing) face limitations in terms of capturing the highly nonlinear interactions and multipath effects characteristic of indoor and urban EM propagation. The complexity increases with the need to model dynamic changes in the environment such as moving objects and varying channel conditions, which are not always well-addressed by conventional approaches (Marey et al., 2022). On the other hand, cGANs (Creswell et al., 2018) are widely used for tasks requiring the generation of new data instances that resemble a given distribution.

## 3 METHODOLOGY

### 3.1 OVERVIEW

In this section, we present our novel approach for augmenting EM ray tracing techniques with a modified cGAN. Our goal is to design a simulator for 3D indoor scenes, the accuracy of which is similar to that of EM ray tracers but is significantly faster for real-time or dynamic scenarios. Figure 1 shows the overall architecture of our network:

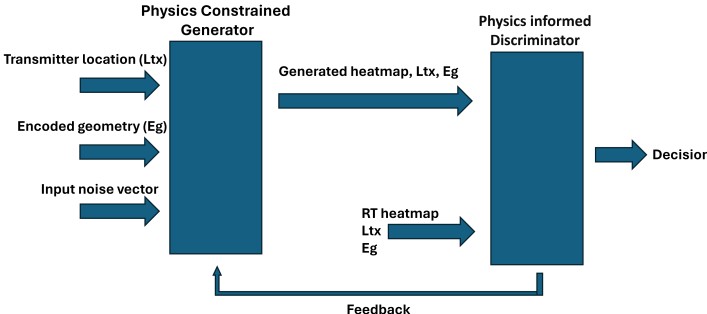

Figure 1: Overall architecture of our cGAN training process. The Generator (G) takes encoded 3D geometry, transmitter location, and a noise vector to output simulated heatmaps. The Discriminator (D) evaluates both the real heatmap from a ray-tracing simulator DCEM and the generated heatmap from G and makes 0/1 decisions.

Our network's formulation can be described as follows:

$$P_r = f_{cGAN}(E_g, L_{tx}, z) \tag{1}$$

where

- $P_r$ is a 3D vector, representing the received power across the indoor environment, as depicted by the generated heatmap (on some height level). This is the primary output of our cGAN model, representing the simulated EM field distribution.
- $f_{cGAN}$ denotes the function computed using the conditional Generative Adversarial Network. It models the complex relationship between the indoor environment's geometry, material properties, the transmitter's location, and the resulting EM signal heatmap.

- $E_{\mathrm{g}}$ represents the encoded geometry information of the indoor environment. It encapsulates details such as the spatial layout in 2D with height information and material properties, which are used for accurate EM propagation modeling.

- $L_{\mathrm{tx}}$ refers to the precise location of the transmitter within the environment. The transmitter's position, in conjunction with the environment's geometry, significantly impacts EM wave propagation and the distribution of received power.

- $z$ represents model or input noise. The noise type is selected through a hyperparameter tuning process. This term accounts for potential discrepancies and uncertainties inherent in the simulation process and is used to improve the accuracy of estimated signal strength throughout the 3D model.

## 3.2 Modified Conditional GAN

In the context of EM simulations, cGANs offer a unique advantage by not only predicting EM field distributions but also generating potential scenarios that could affect these distributions in dynamic environments (Ratnarajah et al., 2022; Kazeminia et al., 2020). We choose cGANs over other learning methods because of several compelling advantages that cGANs offer (Goodfellow et al., 2014), particularly in terms of generating high-quality synthetic data and improving the accuracy and efficiency of path loss predictions. Unlike traditional methods, cGANs are specifically designed for data generation tasks, making them well-suited for creating heat maps based on complex indoor environments and addressing the challenges associated with received power prediction in wireless communication network design and optimization.

Our approach, EM-GANSim, utilizes GANs for several reasons that align with our objectives for real-time and accurate EM simulation in 3D indoor environments:

1. **Data Generation:** GANs are adept at generating synthetic data that closely mirrors the distribution of real data. In the context of EM simulation, this capability allows us to create detailed heatmaps that accurately represent the complex interactions of electromagnetic waves with indoor structures. More details are discussed in the Results and Appendix sections.

2. **Efficiency:** Traditional ray tracing methods can be computationally intensive, especially for large-scale or real-time applications. GANs, once trained, can generate simulations rapidly, which is crucial for achieving real-time performance. A detailed comparison is provided in Sections 5.1 and 5.2.

3. **Flexibility:** GANs can be conditioned on specific parameters, such as geometry and transmitter location, enabling the generation of simulations tailored to particular scenarios without the need for extensive recalculations.

4. **Handling Complexity:** GANs are well-suited to capture indoor EM propagation's highly nonlinear and multipath effects, which can be challenging for conventional simulation methods. For example, conventional ray tracing simulators took much longer time to generate heatmap in average scenes as shown in Section 5.2.

5. **Generalization:** By learning from a diverse dataset, GANs can generalize to new environments and scenarios, which is essential for creating a versatile simulation tool that can be applied across a wide range of conditions. Our method achieves generalization by training the GAN on a diverse dataset of over 2,000 indoor scenes with various room sizes, configurations, and materials.

We present a modified cGAN architecture for our specific task of simulating wireless communication in 3D indoor environments. Our generator takes as input both the geometry information and a noise vector to generate realistic heatmaps that closely match the distribution of the simulated data. The discriminator's role is to distinguish between the real heatmaps derived from EM simulations and the approximate heatmaps generated by the model. We modify the cGAN architecture to account for the EM propagation models to generate accurate heatmaps. Our modified cGAN Error (Generator Network) is defined as:

$$\mathcal{L}_{\mathrm{cGAN}}^{\mathrm{G}} = \mathbb{E}_{E_{\mathrm{g}}, L_{\mathrm{tx}}, z}[\log(1 - D(E_{\mathrm{g}}, L_{\mathrm{tx}}, G(E_{\mathrm{g}}, L_{\mathrm{tx}}, z)))] \tag{2}$$

This equation represents the loss for the generator G in the cGAN and aims to minimize the ability of discriminator D to distinguish generated heatmaps from real ones. The Mean Squared Error (MSE) loss measures the discrepancy between the real received power and the power predicted by the generator, given below:

$$\mathcal{L}_{\text{MSE}} = \mathbb{E}_{E_g, L_{\text{tx}}, P_r}[\|P_r - G(E_g, L_{\text{tx}}, z)\|_2^2] \tag{3}$$

### 3.2.1 GENERATOR

Our generator uses a series of convolutional neural network (CNN) layers designed to capture the intricate spatial relationships within indoor environments. Special attention is given to encoding the geometry information effectively, allowing the model to understand how different materials and layouts affect signal propagation. We also incorporate physical constraints into the objective function, ensuring that the generated samples adhere to the fundamental principles of electromagnetic wave propagation. The generator objective function is given as:

$$\mathcal{L}_{\text{cGAN}}^{\text{G}} = -\mathbb{E}_{E_g, L_{\text{tx}}, z}[\log D(E_g, L_{\text{tx}}, G(E_g, L_{\text{tx}}, z))] + \lambda \mathcal{L}_{\text{MSE}} + \mu \mathcal{L}_{\text{phy}}. \tag{4}$$

This equation combines the cGAN loss with the MSE loss balanced by a weighting factor $\lambda$. Additionally, $\mathcal{L}_{\text{phy}}$ represents the physical constraints loss, and $\mu$ is a weighting factor that balances the importance of the physical constraints in the overall objective function. The physical constraints loss $\mathcal{L}_{\text{phy}}$ includes terms that account for direct propagation, reflection, and diffraction effects:

$$\mathcal{L}_{\text{phy}} = \alpha \mathcal{L}_{\text{direct}} + \beta \mathcal{L}_{\text{reflection}} + \gamma \mathcal{L}_{\text{diffraction}} \tag{5}$$

Where: - $\mathcal{L}_{\text{direct}}$ is the loss due to direct path propagation, calculated as:

$$\mathcal{L}_{\text{direct}} = \sum_{i=1}^{N} \left( PL_d(d_i, f) - \hat{PL}_d(d_i, f) \right)^2 \tag{6}$$

Here, $PL_d(d_i, f)$ is the predicted path loss for direct propagation, and $\hat{PL}_d(d_i, f)$ is the actual path loss based on the formula:

$$PL_d(d, f)[dB] = FSPL(f, d = 1m)[dB] + 10log_{10}(d)[dB] + AT[dB] \tag{7}$$

where $f$ denotes the carrier frequency in GHz, $d$ is the 3D T-R separation distance, $n$ represents the path loss exponent (PLE), and $AT$ is the attenuation term induced by the atmosphere (Okoro et al., 2021). - $\mathcal{L}_{\text{reflection}}$ is the loss due to signal reflections, calculated as:

$$\mathcal{L}_{\text{reflection}} = \sum_{i=1}^{N} \left( PL_r(d_i, f) - \hat{PL}_r(d_i, f) \right)^2 \tag{8}$$

The reflection loss $PL_r$ can be calculated based on reflection coefficients and the geometry of the environment.

- $\mathcal{L}_{\text{diffraction}}$ is the loss due to signal diffraction, calculated as:

$$\mathcal{L}_{\text{diffraction}} = \sum_{i=1}^{N} \left( PL_{diff}(d_i, f) - \hat{PL}_{diff}(d_i, f) \right)^2 \tag{9}$$

The diffraction loss $PL_{diff}$ can be calculated using a modified UTD (diffraction model), which considers the edges and round surfaces of obstacles in the environment (Wang et al., 2024).

By incorporating these physical constraints into the generator's loss function, the GAN is guided to produce outputs that are not only visually convincing to the discriminator but also physically accurate in terms of signal propagation characteristics.

### 3.2.2 DISCRIMINATOR

Our discriminator is also based on CNNs, with the addition of condition layers that incorporate the geometry information. This setup ensures that the discrimination process considers not just the accuracy of the heatmaps but also their consistency with the input geometry. This consistency

refers to a check of the alignment of predicted signal strengths with the expected patterns based on EM propagation theory discussed earlier in the generator, such as maintaining the correct spatial distribution and intensity of signals influenced by environmental factors and material properties. The Discriminator Objective Function is given as:

$$
\begin{aligned}
\mathcal{L}_{\text{cGAN}}^{\text{D}} = & -\mathbb{E}_{E_{\text{g}}, L_{\text{tx}}, P_{\text{r}}}[\log D(E_{\text{g}}, L_{\text{tx}}, P_{\text{r}})] \\
& -\mathbb{E}_{E_{\text{g}}, L_{\text{tx}}, z}[\log(1 - D(E_{\text{g}}, L_{\text{tx}}, G(E_{\text{g}}, L_{\text{tx}}, z)))].
\end{aligned}
\tag{10}
$$

This function models the discriminator's objective, which seeks to identify real and generated heatmaps correctly, thus ensuring that the generated data is accurate

## 3.3 Training

Training of the modified cGAN is performed using a loss function that balances the fidelity of the generated heatmaps as a function of the input geometric conditions. The training process is carefully monitored to prevent mode collapse and ensure a diverse set of realistic outputs. The proposed method is implemented using PyTorch (Paszke et al., 2019) and uses a GPU for efficient model training and inference. For ease of access, we utilize Google Colab, which provides free GPU resources to facilitate the training process. The primary software and dependencies include Python 3 or higher and essential libraries such as NumPy, SciPy, and Matplotlib for data handling and visualization. Our training process on Google Colab takes approximately two days to complete. We optimize the training using hyperparameters such as the learning rate, batch size, and latent space dimensions, which are crucial for achieving the desired model performance and accuracy. A detailed flowchart is presented in the appendix and we plan to release our code at the time of publication.

## 4 Implementation and Performance

We discuss the implementation of our approach and the main issues in terms of obtaining good performance.

## 4.1 Data Adequacy and Quality

Given the complexity of indoor wireless systems, the cGAN would require extensive and high-quality training data that accurately represents the vast array of environmental factors affecting signal propagation. In our process, we generate geometry and power prediction data from WinProp and the DCEM simulator to ensure the diversity and volume of training data, representing different scenarios with high quality.

## 4.2 Hyperparameter Tuning

CGANs are notoriously difficult to train and often sensitive to the choice of hyperparameters, which would require extensive experimentation to fine-tune. In our process, we start with simplified versions of the environment to first train the cGAN before gradually increasing the complexity, which helps the model learn the basic principles before tackling more complex scenarios. By gradually increasing the complexity of the models, we also avoid convergence issues, resulting in a stable solution that provides a realistic simulation of EM ray tracing.

Table 1: Hyperparameters used in GAN Training

| Hyperparameter | Value/Type |
|---|---|
| Learning Rate | 0.0002 |
| Batch Size | 128 |
| Noise Type | Gaussian |
| Loss Function | Binary Cross Entropy |

## 4.3 Mode Collapse

A common issue with cGANs occurs when the generator starts producing a limited range of outputs, which in the case of EM ray tracing could lead to underrepresentation of the solution space. In our work, we included a noise vector in the generator's input to promote feature learning and output variability. A large, diverse training dataset exposed the generator to a range of scenarios, reducing mode collapse risk. Our conditional GAN framework improved output relevance, and adaptive learning rates maintained balanced learning dynamics. These strategies enhanced the model's ability to generate diverse and accurate EM propagation heatmaps.

## 5 Results

This section presents the evaluation of the proposed methodology in terms of accuracy enhancement and efficiency improvement in ray tracing simulations in 3D indoor environments. We have conducted a comparison with WinProp (Jakobus et al., 2018), which is widely recognized as a state-of-the-art solution in EM simulation, as shown in these and more papers (Vaganova et al., 2023) (Wang & Manocha, 2023) (Haron et al., 2021) (Gómez et al., 2023).

We show evaluations in 15 indoor scenes: Scenes 1-15. Detailed specifications of scenes are included in Table 2. On average, the running time of EM-GANSim in any indoor environment is 1 millisecond per data point. However, the models with complex layouts tend to require more computation time than those with single rooms.

Table 2: Detailed Specifications for Various Scenes in terms of size, room configurations, and materials. EM-GANSim is able to predict the signal power strength at any given data location in a few milliseconds.

| Scene# | 1 | 2 | 3 | 4 | 5 |
|---|---|---|---|---|---|
| Type | Multiple rooms | Multiple rooms | Multiple rooms | Single room | Complex floor plan |
| Size ($m^2$) | 25 | 25 | 25 | 144 | 144 |
| Materials used | wood, concrete | wood, concrete, glass | wood, concrete | concrete | wood, concrete, glass |

| Scene# | 6 | 7 | 8 | 9 | 10 |
|---|---|---|---|---|---|
| Type | Complex floor plan | Complex floor plan | Single room | Single room | Single room |
| Size ($m^2$) | 144 | 16 | 16 | 16 | 144 |
| Materials used | wood, concrete, glass | wood, concrete, glass | wood, concrete, glass | concrete, glass | concrete, glass |

| Scene# | 11 | 12 | 13 | 14 | 15 |
|---|---|---|---|---|---|
| Type | Complex floor plan | Multiple rooms | Single room | Single room | Multiple rooms |
| Size ($m^2$) | 144 | 144 | 4 | 16 | 64 |
| Materials used | wood, concrete, glass | wood, concrete, glass | concrete | concrete, glass | wood, concrete, glass |

## 5.1 Accuracy of our Approach

The accuracy of our method is assessed by comparing the simulated received power distributions against standard RT simulations generated using DCEM Wang & Manocha (2022) and WinProp (Jakobus et al., 2018) simulators on our validation datasets in Fig. 2. More heatmap comparisons are shown in the appendix. These heatmaps serve several purposes in supplementing the results: (1) Validation Across Diverse Conditions: These plots demonstrate the model's ability to generalize across different environments by presenting additional scenarios, validating its robustness and adaptability. (2) Comparative Analysis: The plots include comparisons between the EM-GANSim model predictions and those from benchmark WinProp. This comparative analysis highlights the strengths of EM-GANSim in terms of accuracy. (3) Visualization of EM Interactions: The heatmaps visually depict the power distribution and signal propagation across different room layouts. This visualization aids in understanding how well the model captures physical phenomena such as reflection and diffraction.

The comparison underscores the enhanced accuracy achieved by incorporating GAN into the RT simulations, highlighting the advantage of the proposed method in capturing the intricacies of EM wave interactions with indoor structures.

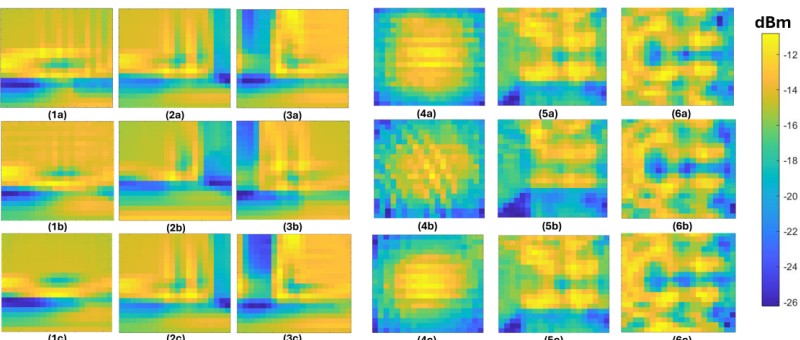

Figure 2: Comparative heatmaps displaying received powers in indoor environments of size 5*5 $m^2$ (left three columns, Scene 1-3) and 12*12 $m^2$ (right three columns, Scene 4-6). First row: WinProp simulation. Second row: GAN-based simulation. Third row: DCEM simulations. The MSEs of GAN-based and DCEM compared to WinProp are shown in Table 4 in the appendix. We see with GAN-based methods that the heatmaps show less MSE in general captures and exhibit more pronounced areas of both high and low signal strength, suggesting a finer granularity in the simulation of received powers.

These are all new scenes not included in the training dataset. The average MSE of GAN-based results of the training set is approximately 3 $dbm^2$ and that of the testing set is around 8.5 $dbm^2$.

In Fig. 3, we show a histogram distribution comparison of the normalized difference in the scenes in the third row of Fig. 2 (Scene 3).

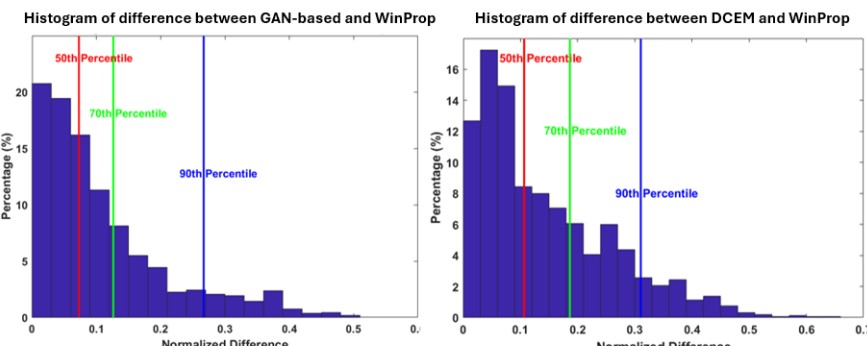

Figure 3: **Left Histogram:** Distribution of the normalized differences in received power levels between the GAN-based simulation and the WinProp simulation for the third-row scene. The vertical lines represent the 50th (median), 70th, and 90th percentiles, indicating a central tendency and spread of the differences. **Right Histogram:** Distribution of the normalized differences in received power levels between the DCEM simulation and WinProp simulation for the third-row scene. The percentiles are marked similarly. We see a tighter distribution in the left graph, suggesting a closer match to WinProp and higher accuracy in the GAN-based simulations.

## 5.2 EFFICIENCY IMPROVEMENT THROUGH GAN

To evaluate the efficiency of using GAN for quick simulations, the computation time was measured and compared between the GAN-based method and the traditional RT approach. The third column in Table 3 is the average generation time of data points in GAN-based predictions, calculated from total time divided by the total number of simulated points. For instance, in a single room of $2m*2m$, with a resolution of $0.05m$, there are 1600 generated data points. Thus, each data point is generated in approximately 2 milliseconds.

Table 3: Computation Time Comparison Between GAN-based and Traditional RT Approaches

|  | GAN-based (seconds) | Traditional RT (seconds) | Generation time per data point (seconds) |
|---|---|---|---|
| **Single room (˜2*2 $m^2$)** | 3.2 | 12 | 0.002 |
| **Multiple rooms (˜8*8 $m^2$)** | 3 | 20 | 0.001 |
| **Complex floor plan (˜12*12 $m^2$)** | 4.3 | 22 | 0.0009 |

The GAN-based method demonstrated a substantial reduction in computation time, offering near-instantaneous simulation results. This efficiency makes the GAN-based approach particularly suitable for applications requiring real-time data analysis and decision-making.

Based on the comparisons after GAN training, we highlight the benefits of GAN below:

- **High-Quality Synthetic Data Generation:** cGANs are adept at generating synthetic data that closely mirrors the distribution of real data, an essential capability for accurately predicting heatmaps from limited real-world data.

- **Efficiency in Prediction:** The GAN-based method can predict heat maps for an entire target area in a single inference step, offering a significant efficiency advantage over traditional, computation-intensive methods.

- **Accuracy Close to Ray Tracing Simulations:** cGANs have the potential to achieve accuracy levels comparable to those of traditional ray tracing simulations by learning to capture the complex variability of path loss across different environments.

## 6 ABLATION EXPERIMENTS

In this section, we first analyze the effect of excluding Gaussian noise from the training process, an element typically introduced to break the symmetry in the model weights, ensuring that different units learn different features. We verify the noise's impact on the generated heatmaps and their respective MSEs. By comparing heatmaps and MSE values, we evaluate the GAN's performance in generating received power distribution in the absence of noise. This ablation study serves not only to reinforce the validity of our methodology but also to offer insights that could refine future implementations of machine learning in EM ray tracing. We define the ablation experiment results as GAN-No-Noise. Observations based on the heatmaps in the appendix, Fig. 7, from the GAN-No-Noise, GAN-based, and DCEM predictions in testing scenes are discussed as follows:

- **First Row (GAN-No-Noise):** The absence of Gaussian noise results in less varied and uniform heatmaps, indicating potential over-smoothing and reduced accuracy in capturing EM wave interactions within the environment.

- **Second Row (GAN-based with Noise):** Inclusion of noise introduces more defined contrasts and a broader range of power levels, suggesting a better representation of the complex nature of EM propagation and environmental features.

- **Cons of GAN-No-Noise:** Lack of noise in training leads to simpler patterns, reduced model accuracy, and potential issues in generalizing to new environments, which is critical for applications like network planning.

- **Importance of Noise:** Gaussian noise is essential in training to break symmetry in the model, ensuring diverse learning and preventing the network from collapsing into repetitive pattern production.

These observations underline the importance of including noise in the GAN training process to enhance the model's ability to predict received power distributions accurately and robustly, especially when applied to complex indoor EM propagation scenarios.

We also include the corresponding MSE of GAN-No-Noise and GAN-based compared to DCEM in Table 5 in the appendix. For these three testing cases, GAN-based predictions consistently have lower MSE values than GAN-No-Noise, indicating that the inclusion of noise during the training process contributes to a more accurate prediction of received power levels. The improved MSE with noise suggests that Gaussian noise acts as a regularizer, preventing the model from memorizing the training data and instead forcing it to learn the underlying distribution. The presence of noise

also introduces a wider variety of scenarios during training, making the GAN model more robust to unseen environments and better at generalizing from the training data.

Another ablation test is designed to evaluate the impact of incorporating physical constraints into the objective function of our generator, included in the appendix. These physical constraints are integrated to ensure that the generated samples adhere to the fundamental principles of electromagnetic wave propagation, accounting for direct propagation, reflection, and diffraction effects. The ablation tests will involve running the simulator under two distinct conditions:

- **With Physics Constraints**: The generator's objective function will include the physical constraints loss ($\mathcal{L}_{phy}$), which comprises terms for direct path propagation ($\mathcal{L}_{direct}$), reflections ($\mathcal{L}_{ref}$), and diffractions ($\mathcal{L}_{diff}$).
- **Without Physics Constraints**: The physical constraints loss ($\mathcal{L}_{phy}$) will be omitted from the objective function, leaving only the cGAN loss and the MSE loss components.

This ablation test demonstrates the impact of incorporating physical constraints by comparing the performance and accuracy of the generator under both conditions. Key performance metrics observed were:

- **Signal Propagation Accuracy**: The tests revealed that the generator with physical constraints produced more accurate signal propagation characteristics. The predicted path losses (direct, reflection, and diffraction) closely matched the actual path losses, highlighting the effectiveness of the constraints in capturing the physical phenomena of EM wave propagation in indoor environments.
- **Visual and Structural Fidelity**: The generated samples with physical constraints exhibited higher visual realism and structural coherence. These samples were more accurate in modeling the indoor environments compared to those generated without the constraints.

## 7 Conclusions, Limitations, and Future Work

We present a novel approach that uses ML methods along with EM ray tracing to enhance the accuracy and efficiency of wireless communication simulation within 3D indoor environments. We use a modified cGAN that utilizes encoded geometry and transmitter location and can be used for accurate EM wave propagation. We have evaluated its performance on a large number of complex 3D indoor scenes and its performance is comparable to EM ray tracing-based simulations. Furthermore, we observe a 5X performance improvement over prior methods.

Our study enhances wireless communication efficiency and lays the ground for future real-time applications. Our approach has some limitations. Since our training data is based on ray tracing, our prediction scheme may not be able to accurately model low-frequency or other wave interactions. Our current approach is limited to indoor scenes, and we would also like to evaluate it in scenes with multiple dynamic objects. A key challenge is to extend and use these methods for large urban scenes with complex traffic patterns to model wireless signals.

Furthermore, we plan to add dynamic elements, such as movable partitions and furniture, to simulate real-world changes in indoor layouts. By leveraging publicly available architectural data (such as the 3D-Front dataset (Fu et al., 2021)), we will continuously update the dataset with new scenarios that reflect emerging trends in building design and technology. This comprehensive dataset expansion will improve the model's ability to predict EM wave propagation in complex and varied indoor environments, ultimately enhancing its applicability and reliability in practical applications.

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

## A APPENDIX

Table 4: MSE of GAN-based and DCEM compared to WinProp

|  | **GAN-based** ($dbm^2$) | **DCEM**($dbm^2$) |
|---|---|---|
| **Scene 1** | 7.29 | 5.60 |
| **Scene 2** | 9.47 | 9.08 |
| **Scene 3** | 8.51 | 11.00 |
| **Scene 4** | 12.03 | 6.42 |
| **Scene 5** | 11.71 | 9.44 |
| **Scene 6** | 5.91 | 7.36 |
| **Scene 7** | 7.66 | 10.93 |
| **Scene 8** | 7.93 | 4.47 |
| **Scene 9** | 9.76 | 7.95 |
| **Scene 10** | 8.35 | 6.72 |
| **Scene 11** | 8.67 | 8.61 |
| **Scene 12** | 6.94 | 7.12 |

Fig. 4: sample 3D renderings of indoor environments used in the training set.

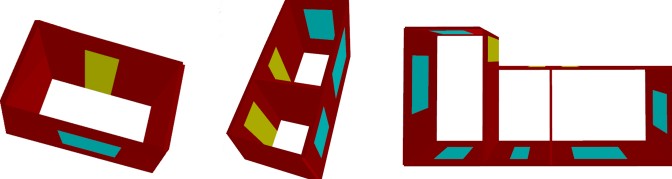

Figure 4: Sample 3D renderings of indoor environments used for simulation: (a) Single-room setup with minimal furniture. (b) Multi-room configuration with complex wall structures. (c) Multi-room layout with varied dimensions and partitions. These scenes demonstrate the diversity of layouts the ML model must interpret for accurate EM ray tracing simulation. The red represents concert walls, the blue represents glass, and the yellow represents wooden doors.

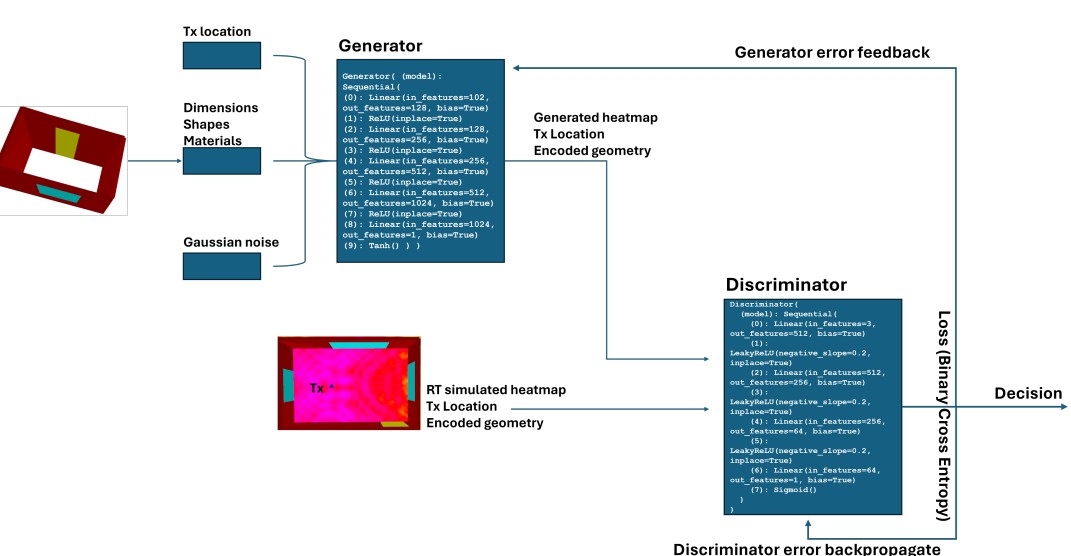

Figure 5: A more detailed flowchart of the GAN training process and implementation details: After data preparation, we encode geometry info along with transmitter location and a noise vector to feed into the generator networks. The generator employs a series of convolutional neural network (CNN) layers designed to capture the intricate spatial relationships within the indoor environments. Special attention is given to geometry information, allowing the model to understand how different materials and layouts affect signal propagation. The discriminator is also based on CNNs, with the addition of condition layers that incorporate geometry information. This setup ensures that the discrimination process considers not just the realism of the heatmaps but also their consistency with the input geometry. The loss function is selected as binary cross-entropy, backpropagated through the respective networks to compute the gradient of the loss with respect to the network weights. Gradient descent optimization algorithms are used to adjust the weights of the generator and discriminator in the direction that will reduce their respective losses.

Fig. 7: detailed flowchart of the GAN training process and implementation details.

Fig. 6 shows more prediction accuracy comparison of WinProp, DCEM and GAN-Based results. We see that GAN-based tends to have a larger received power MSE than DCEM, which suggests some accuracy degradation while achieving the fastest running time among other methods.

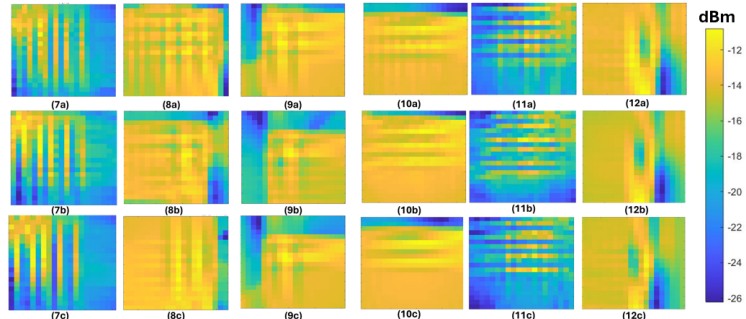

Figure 6: Comparative heatmaps displaying received powers in indoor environments. First row: WinProp simulation. Second row: GAN-based simulation. Third row: DCEM simulations. The room sizes on the right are larger than those on the left.

Fig. 7 shows noise ablation test heatmap comparisons.

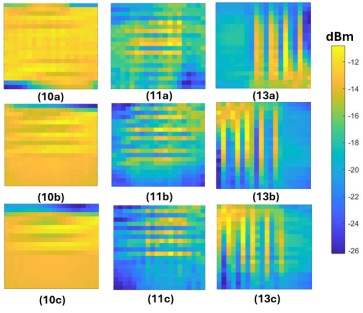

Figure 7: Comparative heatmaps displaying received powers in indoor environments. First row: GAN-No-Noise. Second row: GAN-based with noise. Third row: DCEM predictions (benchmark). We use the DCEM results as the benchmark and compare the results from GAN-No-Noise and GAN-based.

Table 5: MSE of GAN-No-Noise and GAN-Based compared to DCEM

|          | GAN-No-Noise ($dbm^2$) | GAN-based($dbm^2$) |
|----------|------------------------|--------------------|
| Scene 10 | 10.88                  | 5.24               |
| Scene 11 | 9.52                   | 7.19               |
| Scene 13 | 16.38                  | 3.65               |

Fig. 8 shows physics-constrained ablation test heatmap comparisons. This comparison highlights the crucial role of physical constraints in enhancing the accuracy and realism of the GAN-based model for simulating indoor signal propagation, as evidenced by the closer alignment with the DCEM benchmark.

Fig. 9 aims to show the robustness and generalization of our EM-GANSim approach across diverse conditions. The CAD models used to generate these plots are derived from a dataset of 3D indoor environments, which is discussed in Section 3.2. These models are selected to reflect the complexity and diversity of real-world indoor environments. This complexity arises from several factors: (1) Varied Room Configurations: The models include multiple room layouts with different sizes and

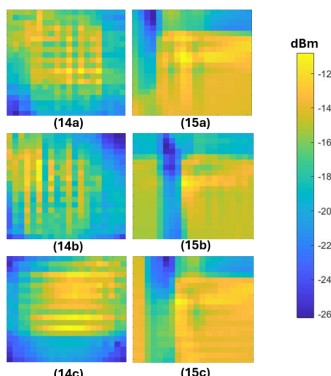

Figure 8: Comparative heatmaps displaying received powers in indoor environments. **First row:** DCEM predictions (benchmark). These heatmaps represent the received power as predicted by DCEM, serving as a benchmark for comparison. The spatial distribution of received power follows expected patterns based on the known physical principles of EM wave propagation. **Second row:** GAN-based with physics constraints. The heatmaps show the predictions from the GAN model where physical constraints have been incorporated into the objective function. These results closely align with the benchmark predictions, indicating that the inclusion of physical constraints helps the model adhere to the fundamental principles of signal propagation, capturing direct propagation, reflections, and diffraction effects accurately. **Third row:** GAN-based without physics constraints. These heatmaps represent the predictions from the GAN model without physical constraints in the objective function. The spatial distribution of received power deviates from the benchmark predictions, demonstrating the model's struggle to accurately capture the complex interactions in signal propagation without the guidance of physical constraints. The absence of physics-based loss terms results in less realistic and less reliable predictions.

shapes, ranging from simple square rooms to intricate floor plans with interconnected spaces and corridors. (2) Material Diversity: The inclusion of diverse materials like concrete, wood, and glass helps simulate the varying reflective, absorptive, and diffractive properties found in actual buildings. (3) Obstacles and Furnishings: The models feature obstacles such as walls and partitions, which affect EM wave propagation through reflection, diffraction, and scattering. The first row demonstrates results from a benchmark method from WinProp for comparison. The second row of plots represents predictions from the EM-GANSim model, showcasing its capability to accurately predict electromagnetic wave interactions in various indoor environments.

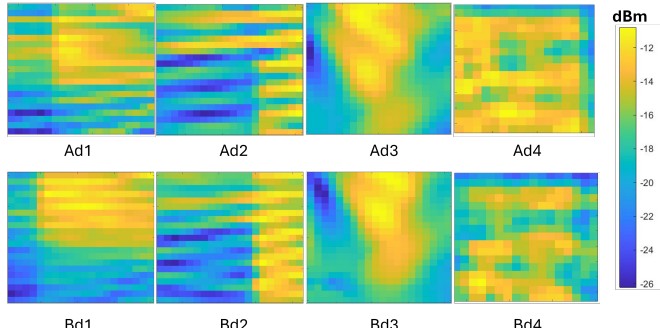

**First row:** WinProp simulation (benchmark). **Second row:** GAN-based simulation.
This plot is to answer the Question 5 from Reviewer EZUU

Figure 9: First row: Winprop simulations (benchmark); Second row: GAN-based simulations, showcasing its capability to accurately predict electromagnetic wave interactions in various indoor environments

