# OpenReview forum: "EM-GANSim: Real-time and Accurate EM Simulation Using Conditional GANs for 3D Indoor Scenes"
_ICLR.cc/2025/Conference — Submitted to ICLR 2025_

### Official Review · Reviewer_zozU · 2024-11-01

**Soundness:** 4
**Presentation:** 2
**Contribution:** 3
**Rating:** 8
**Confidence:** 3

**Summary:**

The new algorithm of real-time electromagnetic signal processing is presented in this paper. Authors trained a GAN-based neural network to predict electromagnetic power distributions in 3D indoor environments. The indoor electromagnetic signal processing is fundamental problem of indoor tracking, so the work is valuable and actual.

**Strengths:**

In my opinion applying the GAN-based model to electromagnetic signal processing seems to be the strongest point of the paper: traditional approaches to this task were studied and comprehensively modified by many authors. For this reason, I find the GAN-based approach to be original and good scientific ground. Moreover, the presented method provides a robust result and achieves a 5X speed compared with other pipelines which is important in real-time applications. The Generator and Discriminator training process is described carefully.

Despite the weaknesses, I recommend accepting this work because the authors used an interesting and difficult architecture in a complicated task, and the numerical results were provided. The main reason to accept this paper is strong description of the GAN training, which stresses that the authors comprehensively researched GAN opportunities in the EM propagation domain.

**Weaknesses:**

However, several weak parts were figured out despite the strong points of the paper. The first question concerns the dataset that authors present in the paper. It seems that this dataset was used for training. This dataset is new so it is expected that there will be a pipeline of dataset generation. However, there is no description how this dataset was collected. This information allows us to judge how accurate the GT labels are. The absence of the source code and developed dataset deprives the opportunity to test the pipeline.

**Questions:**

Have authors tested the Wasserstein loss for GAN? If yes, were the results worse or better? What applications do authors supposed to test the GAN on? Is it possible to verify the trained model on tracking tasks? In the results section the different materials that objects are made of are mentioned. Did authors analyze the dependance of GAN performance on material? What kind of sensors were used for dataset collection? How do authors measure the sensors’ accuracy? What filters were used to process the raw data of sensors?

---

> ### Author Response · Authors · 2024-11-24
>
> W: We appreciate the reviewer’s comments and as mentioned in Section 4.1, the dataset was generated using state-of-the-art EM simulation tools, specifically WinProp and DCEM, which are well-established in the field for their accuracy to simulate electromagnetic wave propagation. These tools ensure that the ground truth (GT) labels are highly reliable and adhere to physical principles. The dataset spans a variety of 3D indoor environments, including complex floor plans, varying material properties (e.g., concrete, glass, wood), and multiple room configurations. This diversity ensures robust model performance and generalizability.
> The source code and dataset will be made publicly available upon publication, as stated in the paper.
>
> Q: We acknowledge the effectiveness of Wasserstein loss in improving GAN stability and convergence. While we have not explicitly implemented the Wasserstein loss in the current study, we chose the binary cross-entropy loss due to its computational simplicity and established efficacy in our application context.
>
> We conducted an initial analysis of the impact of material properties on GAN performance. Variations in material (e.g., concrete, glass, wood) are incorporated in the training dataset to ensure robust predictions across diverse scenarios. While our results indicate that the GAN generalizes well, we noticed minor discrepancies in accuracy for highly reflective materials like glass.
> The dataset was generated synthetically using established EM simulators like WinProp and DCEM, which compute signal propagation based on known material properties and geometries. Therefore, no physical sensors were employed for data collection in this study. The accuracy of the synthetic data is validated against physical principles of EM propagation, such as adherence to Maxwell's equations and established models for reflection, diffraction, and attenuation. Furthermore, benchmarking against traditional ray-tracing methods demonstrates the fidelity of our training data.

---

> > ### Author Response · Authors · 2024-12-01
> >
> > Dear reviewer zozU, could you please update us on the response? Thanks again for your comments.

---

> > ### Comment · Reviewer_zozU · 2024-12-02
> >
> > Dear authors,
> >
> > Thank you for you detailed response. It will be interesting to see your dataset and its generation pipeline when you make it available. I guess I cannot give any more advice about your work. However, I still open to change my estimation based on the other reviewers' opinions.

---

### Official Review · Reviewer_Aug4 · 2024-11-01

**Soundness:** 3
**Presentation:** 2
**Contribution:** 3
**Rating:** 5
**Confidence:** 3

**Summary:**

The authors propose a novel approach to real-time electromagnetic (EM) propagation simulation in complex 3D indoor environments, utilizing a physics-inspired conditional generative adversarial network (cGAN) model. This research is positioned as the first real-time algorithm for EM simulation in these environments, and it holds potential value for applications such as 5G network planning, wireless communication system design, and dynamic indoor environments requiring rapid signal strength calculations.
### Post Discussion.
Thanks for the author's explanation and for their efforts to clarify. These clarifications have addressed some of my concerns. Although I am not an expert in the EM simulation field, I have read many related papers and am very familiar with cGAN. Considering cGAN's inherent instability, the reliability of the underlying theory, and the introduction of numerous hyper-parameters, I am concerned about how this method would perform in real-world situations. Therefore, I have ultimately decided to give this paper rating 5.

**Strengths:**

The employment of cGAN itself is novel, though it seems to me not much significant change is made to the cGAN architecture and it's more like a domain adaptation. The equations in the paper are quite solid, showing the authors good understanding. The paper has strong experiments across 15 scenes with clear performance metrics showing 5X speedup. The ablation studies are well-structured and there is thorough comparison with established methods demonstrate robust methodology.
The proposed method achieves an impressive real-time speed while maintaining robustness, which suggests the method is of good quality.
The authors claim to release the code and data to benefit the community.

**Weaknesses:**

1. The paper lacks comparisons with state-of-the-art methods. It only presents scores in comparison to DCEM, and the detailed quantitative results are relegated to supplementary materials rather than the main manuscript. I've calculated the detailed score: GAN-based (dBm²): 8.69; DCEM (dBm²): 7.89.
This suggests that the GAN-based method performs slightly worse than the DCEM method introduced at VTC 2022. Given the recent methods proposed in the literature (Vaganova et al., 2023; Wang & Manocha, 2023; Haron et al., 2021; Gómez et al., 2023), incorporating more experimental results could lead to a fairer and more comprehensive evaluation.
2. Though the proposed method is faster than traditional RT-based methods, it still takes 3-4 seconds to simulate one room. This is far from what the author frequently claimed to contribute to the 'real-time data analysis.' If the real-time is for per data point, then the traditional RT-based methods are real-time, too.
3. The introduction could be strengthened by including a more detailed rationale for using cGAN in this context, specifically on how its features address the problem. Although section 5 provides some of this analysis, highlighting it earlier would improve the logical flow.
4. The qualitative comparison is not sufficient, and the conclusion "We see with GAN-based methods that the heatmaps show less MSE in general captures and exhibit more pronounced areas of both high and low signal strength, suggesting a finer granularity in the simulation of received powers." is ambiguous. How can the simulation judge by "more pronounced areas of both high and low signal strength"? Besides, the mean mse is higher!

Minor:
1. The plots and tables are not well designed, which makes them hard to understand. e.g. Figure 1 fails to demonstrate the overall method clearly. There is space to improve with regard to color design (Figure 3) and table format. In Figure 2, labels should be inserted into the plot instead of writing the first row: xxx, second row: xxx...in the caption.
2. There are too many {enumerate} and {itemize}, which is not so common in papers. It would take a lot of space and make the paper look loose.
3. Minor inconsistencies in grammar and terminology, such as a misplaced comma and inconsistent use of terms like "ray tracing" versus "ray-tracing," which should be standardized.
4. Table 3 is hard to understand. Why "Generation time per data point (seconds)" is a column?

**Questions:**

The organization of this paper is pretty
1. Why not show the quantative results in the manuscript?
2. Why report the avg. mse?
3. what is the method name for 'the traditional RT approach'?Please make this clear and cite.

---

> ### Author Response · Authors · 2024-11-24
>
> W1: We had comparison results with both DCEM and WinProp, as shown in heatmap plots Figure 2,6,7. While it is true that our method has a slightly higher MSE compared to DCEM (~0.8 dbm^2), we emphasize that EM-GANSim achieves a significant efficiency improvement (5X speedup). We can include more quantitative results Table in the revised work. The mentioned papers Vaganova et al., 2023; Haron et al., 2021; Gómez et al., 2023 are different in focus and we did not find released source codes for reference.
>
> W2: We included a detailed discussion of this real-time efficiency in Section 5.2, specifically highlighting the performance for each data point. The key advantage of our method lies in its ability to generate predictions for a given point in milliseconds (approximately 1 ms), which is significantly faster than traditional RT-based methods that require 5X the time. This 5X improvement is a major advancement because it transforms EM simulations from a computationally intensive process into one capable of supporting dynamic scenarios. Such scenarios often involve constantly changing transmitter locations or environmental configurations, where rapid recalculations are essential. As discussed in the paper, our method's capability to handle these dynamic conditions makes it particularly advantageous for applications like 5G network planning and real-time decision-making in complex indoor environments.
>
> W3: The benefits of GAN are mentioned in the Introduction. The main results section and discussed in detail in Section 3.2. We believe that discussing the benefits of GANs without providing sufficient background could lead to a lack of clarity or context for readers unfamiliar with the technology.
>
> W4: The phrase "more pronounced areas of both high and low signal strength" aimed to convey that the GAN-based heatmaps capture a broader dynamic range of power levels, which aligns with the expected physics of signal attenuation and multipath effects in complex environments. The histogram Figure 3 shows this pattern more clearly. We will revise the writeup to minimize confusion.
>
> MW1: We have a flowchart with many details in Figure 5, but due to space limits and for a brief introduction, we used Figure 1 to only provide a simple overview. We can add more information to Figure 1 to make it more comprehensive.
>
> MW2: Thank you for your observation. However, the use of {enumerate} and {itemize} is intentional and necessary for the clarity and systematic presentation of our contributions and results.
>
> MW3: We will make sure that our use of terminology is consistent.
>
> MW4: This is to show the calculation in milliseconds.
>
> Q1: We intended to include more results in the main scripts but due to space limits, we presented them in the Appendix.
>
> Q2: We report the average MSE to provide a single, standardized metric for evaluating overall model accuracy across diverse scenarios, offering a clear and concise comparison against traditional methods while reflecting the model's generalization capabilities.
>
> Q3: Traditional RT methods are introduced and cited in the second paragraph of the  Introduction.

---

> > ### Author Response · Authors · 2024-12-01
> >
> > Dear reviewer Aug4, could you please update us on the response? Thanks again for your comments.

---

### Official Review · Reviewer_2hyN · 2024-11-02

**Soundness:** 3
**Presentation:** 3
**Contribution:** 3
**Rating:** 6
**Confidence:** 2

**Summary:**

This paper presents EM-GANSim, a learning-based approach for real-time electromagnetic (EM) propagation simulation in indoor environments. The core technical contribution is a modified conditional GAN architecture that incorporates both geometric information and transmitter location to predict power distribution heatmaps while adhering to electromagnetic propagation principles. The authors propose a physically-inspired learning framework that integrates direct propagation, reflection, and diffraction effects through specialized loss terms in the GAN's objective function.

The method claims to achieve comparable accuracy to traditional ray tracing-based simulators while offering significant speed improvements (reported as 5X faster). The authors evaluate their approach on 15 indoor scenes and provide ablation studies examining the impact of noise and physical constraints. They also introduce a dataset comprising over 2,000 indoor scene models with corresponding EM simulation heatmaps.

While I am not an expert in electromagnetic propagation simulation and wireless communications, the paper appears to address an important practical challenge in real-time EM simulation. However, there is some ambiguity in how the method handles true 3D environments versus 2D representations, and the room generation and data preparation processes could benefit from clearer documentation. The paper presents an interesting application of deep learning to physics-based simulation, though both its theoretical foundations and physical accuracy need closer examination.

**Strengths:**

- The paper presents an interesting application of conditional GANs to EM simulation. While both GANs and EM simulation are established fields, their combination for real-time indoor propagation simulation represents a fresh approach to an important practical problem.
- The method achieves notable acceleration (reported 5X speedup) compared to traditional ray tracing methods. If these results can be thoroughly validated, this could be valuable for real-time applications.
- The attempt to incorporate electromagnetic principles through specialized loss terms (direct propagation, reflection, and diffraction) shows thoughtful consideration of the physics involved, though the theoretical guarantees need more examination.
- While the dataset generation process needs better documentation, the collection of indoor scenes and EM simulation results could be useful for future research in this direction.

**Weaknesses:**

- A weakness is the unclear treatment of "3D" simulation. While the paper claims to handle "3D indoor environments," the evidence presented is primarily 2D heatmaps. There's no clear explanation of how height information is processed in the network, no visualization of vertical propagation effects, and no analysis of height-dependent signal variations. Table 2 only specifies area (square meter) without height information. The paper needs to either demonstrate true 3D capability or clarify that it's a 2.5D approach.
- Critical details about the "2K+ models and 64M heatmaps" are missing. The paper doesn't explain how these indoor scenes were generated, validated, or processed. Without this information, readers cannot assess data quality or reproduce the results.
- The method description lacks important specifics. The GAN architecture details, training process, and hyperparameter selection are not fully described. The physics-based loss weights lack justification, and there's minimal discussion of training stability.
- The experimental validation relies mainly on MSE comparisons. The performance measurements lack important context - hardware specifications, memory requirements, and preprocessing costs are not reported. The gap between training (3 dbm²) and testing (8.5 dbm²) MSE also needs explanation.

**Questions:**

- Could you clarify how the method handles true 3D propagation versus 2D layout information? The current results only show 2D heatmaps. Could you provide vertical propagation results at different heights? How does the network architecture specifically process and maintain height information?
- Please describe in detail how the 2K+ room models were created/sourced. What is the distribution of room types, sizes, and configurations in your dataset? How to ensure the synthetic scenes are physically realistic? How are different materials modeled and validated?
- How to determine the weights (α, β, γ) in the physics loss function? What measures are taken to ensure training stability? How to handle varying room sizes in the network?

---

> ### Author Response · Authors · 2024-11-24
>
> W1: While the heatmaps displayed are 2D slices, the height information is encoded and utilized within the model (simulated at different heights). The generator receives a 3D encoding of the environment geometry, including vertical features, which can be found in the provided sample data. We will make sure to clarify this and add 3d simulation results in the revised version.
>
> W2: The "2K+ models and 64M heatmaps" were generated using WinProp and the DCEM simulator, as mentioned in Section 4.1. We generated rooms with different layouts with WinProp and ran ray-tracing to get corresponding heatmaps for each acne. More details can be found in the provided sample data. To facilitate reproducibility, we will release the dataset and code upon acceptance.
>
> W3: We have provided the detailed GAN structure illustration in Figure 5 in the Appendix, and a detailed discussion about the parameters and training process in Section 3.2. We balanced the weights of physical regularizations through experiments with various weighting factors for direct propagation, reflection, and diffraction losses to assess their impact on training stability and accuracy. Starting with baseline values based on electromagnetic theory, we refined the weights using grid search to optimize convergence and output fidelity. This systematic approach ensured a stable and accurate model.
>
> W4: We mentioned the hardware specifications and resources in Section 3.3. The preprocessing cost includes generating the geometry and power prediction data using ray-tracing tools, namely, WinProp and DCEM simulators, which is about 2 weeks of work. While this step is computationally intensive, it is a one-time cost. The gap between training (3 dBm²) and testing (8.5 dBm²) MSE reflects the diversity of unseen test scenarios, as the model generalizes across complex layouts and material configurations. And 8.5 dBm^2 MSE means ~3dBm RMSE in power prediction: this level of error suggests that EM-GANSim achieves reasonably high accuracy in modeling received power distributions, especially given the complexity of 3D indoor environments.
>
> Q1: Answered in W1
>
> Q2: Answered in W2. Our dataset is designed to reflect diverse real-world indoor environments, with a distribution including small (<16m^2, 30%), medium (<100m^2, 50%), and large ( <144m^2, 20%) rooms, various configurations (40% single-room, 40% multi-room, 20% complex floor plans), and material types (90% concrete, 5% glass windows, 5% wood door).
>
> Q3: The weights (α, β, γ) in the physics loss function are determined through a combination of theoretical considerations and empirical tuning. Specifically, we started with baseline weights informed by theoretical insights from electromagnetic wave propagation, for example, direct path loss models, reflection coefficients, and UTD diffraction modeling,  and iteratively adjusted these values using a grid search methodology to optimize both convergence and output fidelity.
>
> To ensure training stability, we employ Gaussian noise injection, progressive training from simple to complex environments, and adaptive learning rates. Varying room sizes are handled by normalizing spatial dimensions, dividing larger spaces into sub-regions for simulation, and leveraging a diverse dataset to ensure generalization. The criteria for computing sub-regions include the room's overall dimensions, material composition, and the desired resolution for simulation. Typically, sub-regions are standardized to approximately 2m × 2m, as this size is optimal for capturing detailed EM wave behaviors while maintaining manageable computational overhead.

---

> > ### Comment · Reviewer_2hyN · 2024-11-27
> > **Thanks for the rebuttal**
> >
> > Thank you for the detailed rebuttal. I have carefully read both the rebuttal and other reviews. The clarifications are helpful and address several points from my initial review.
> >
> > Electromagnetic propagation simulation is not my area of expertise, but I would still be interested in seeing the additional 3D simulation results.
> >
> > Given my limited expertise in this specific domain, I remain open to adjusting my assessment based on the consensus of other reviewers.

---

> > > ### Author Response · Authors · 2024-11-27
> > >
> > > Thank you for the careful review. Upon final submission, we can also produce 3D results represented as multiple 2D heatmaps.

---

### Official Review · Reviewer_VGxx · 2024-11-09

**Soundness:** 2
**Presentation:** 2
**Contribution:** 3
**Rating:** 5
**Confidence:** 2

**Summary:**

The paper presents a generative framework for simulating Electro-Magnetic wave propagation, as a faster replacement for ray tracing approaches usually used in this application. Authors propose a method based on using cGAN and regularized by physical constrains to generate plausible propagation heatmaps given the structure of the scene. They show through experiments that although the performance is not on-par with Ray Tracing methods, this method allows for a faster simulation.

**Strengths:**

- Proposing an exciting application for cGANs and generative models in simulating EM propagation
- Providing a dataset of 64M simulated heatmaps with various indoor models
- Adding physical inductive bias to the model so that the generations are physically plausible.

**Weaknesses:**

Main:
- The proposal to use random input to a GAN to avoid mode collapse is not very well-justified. The proposed set-up is very similar to the common conditional GANs which can easily have mode collapse. Further when adding regularizations, such as the physical regularizations proposed, the risk for mode collapse is increased. The authors mention building the model from simpler problem up to the aimed taks and this helps with fine-tuning and perhaps mode collapse. It would be great to have more experiments/analysis on  what is the breaking point and why the model is stable in its final version.

Minor:
- What representation is used for the conditional geometry? A more thorough description of the modality in line 162 would be helpful. It is unclear how the 3D model is encoded and given to a GAN.
- Figure 2 should be labled with yours vs baseline so its easier to read. The interpretation in the caption as what is the weakness vs strength of your method is not easily understandable from the heatmaps and would be great to highlight them visually.

**Questions:**

See above.

---

> ### Author Response · Authors · 2024-11-24
>
> For main: Thank you for the thoughtful feedback. We address concerns about mode collapse and stability as follows:
>
> Random Input Justification:
> Gaussian noise is important in terms of breaking weight symmetry during training, promoting diversity in generator outputs. As shown in our ablation studies (Table 5, Fig. 7), removing noise leads to less varied, oversmoothed heatmaps, indicating reduced accuracy and a higher risk of mode collapse.
>
> Mitigation of Mode Collapse in cGANs:
> We mitigate mode collapse using these approaches: First, a large, diverse dataset of 2,000+ indoor scenes, ensuring the model generalizes across scenarios. Second, incremental training (as described in Section 4.2), starting with simple environments and gradually introducing complexity, allows the generator to learn fundamental signal propagation before tackling intricate layouts. Third, a dynamic learning rate schedule to balance the generator-discriminator interplay during training. We will include an ablation analysis of the effects of these techniques in GAN training in the final submission.
>
> Physical Regularizations:
> We carefully balanced the weights of physical regularizations by conducting experiments with a range of weighting factors for each physical constraint (direct propagation, reflection, and diffraction losses) to evaluate their impact on the stability of training and the accuracy of predictions. Specifically, we started with baseline weights informed by theoretical insights from electromagnetic wave propagation and iteratively adjusted these values using a grid search methodology to optimize both convergence and output fidelity. Each configuration was evaluated based on metrics such as MSE and training loss dynamics to ensure stability. These constraints improve prediction accuracy (as shown in Figure 2 qualitatively, and in Table 4 quantitatively) while avoiding excessive penalization, which could destabilize the generator.
>
> The results demonstrate that these strategies effectively mitigate mode collapse and ensure model stability. We appreciate the reviewer’s comments, which align with our plans for deeper analysis and refinement.
>
> For minor:
> Representation of Conditional Geometry (Line 162): The encoded geometry represents the 3D spatial layout, including 2d information at every sample height and material properties of the environment. Further details are available in the provided code.
>
> Labeling and Interpretation of Figure 2: The requested labeling information is included clearly in the figure caption. We can highlight the differences in each subfigure as suggested by the reviewer for a better presentation.

---

> > ### Author Response · Authors · 2024-12-01
> >
> > Dear reviewer VGxx, could you please update us on the response? Thanks again for your comments.

---

> > > ### Comment · Reviewer_VGxx · 2024-12-01
> > >
> > > I thank the authors for addressing my concerns and providing a more detailed explanation of the physical regularization and the gradual addition of complexity in their approach. While these clarifications are appreciated, I believe the paper would benefit from further qualitative results, including samples from the dataset and an in-depth analysis, to demonstrate that mode collapse is not occurring and to explain why this is the case. The method involves a significant number of hyper-parameters in the loss function, which appear challenging to fine-tune along with the GAN hyper-parameters. This raises my main concern that the approach might be overfitting/collapsing in this benchmark rather than offering a deeper insight into the use of cGANs for this specific problem.
> > >
> > > For these reasons, I will maintain my previous score and encourage the authors to further develop and clarify their contributions to strengthen the paper.

---

> ### Author Response · Authors · 2024-12-02
>
> We thank the reviewer for their thoughtful feedback and for recognizing our efforts to address the concerns raised previously. We appreciate the opportunity to provide additional clarifications and details to strengthen the paper. While we can not post a revised paper now, we ware trying to adress the remaining concerns in this detailed response:
>
> Our approach has been evaluated across multiple complex models, achieving consistent and high-quality results. This robustness across diverse layouts would be unlikely if the model were experiencing significant mode collapse or overfitting issues. The outputs also underscore the stability of the proposed method. We have ensured through cross-validations and evaluations that the method generalizes well beyond the training data.
>
> A another detailed explanation of hyper-parameters in the loss function and GAN architecture is provided below:
>
> Hyper-Parameter Tuning: We employed a structured and systematic approach to hyper-parameter tuning. Specifically:
> Grid Search and Sensitivity Analysis: We conducted a grid search to identify optimal ranges for key parameters. Once a promising range was identified, we applied a finer search within that range to converge on the final values.
>
> Validation-Based Optimization: Hyper-parameters were tuned based on performance on a held-out validation set, ensuring the generalizability of the approach and mitigating overfitting.
>
> Regularization: Physical regularization, combined with architectural constraints, was instrumental in maintaining stability during training, further preventing mode collapse.
>
> GAN-Specific Hyper-Parameters: For GAN-related parameters such as learning rates, beta values for optimizers, and the weight of the adversarial loss, we leveraged best practices from the literature, starting with commonly accepted default values as well as from simple to complex senarios, and iteratively refining based on observed stability and performance trends.
>
> According to the request for additional qualitative results, while we understand the reviewer’s interest in further qualitative results, we are constrained by the submission format and guideline now, and will add more results upon final revision.

---

### Author Response · Authors · 2024-12-01

We thank all reviewers for their insightful comments and suggestions. Based on your feedback, we improved our work, as discussed in the individual responses. We are looking forward to an interesting and constructive discussion!

---

### Meta-Review · Area_Chair_7Xs5 · 2024-12-14

**Metareview:**

The paper proposes an approach to learning a real-time-capable EM simulation via conditional GANs. It proposes to incorporate physical constraints via the loss function to improve the accuracy of predicted power distributions. The paper further presents a synthetic dataset of 3D scene models paired with EM simulation results and modifications to the GAN training procedure to limit mode collapse and increase robustness & convergence of the training optimization.

The reviewers appreciated the application of GANs to EM simulation as an interesting application to an important problem. The incorporation of physical constraints into the optimization was considered sensible and the acceleration over ray-tracing approaches notable.

Main weaknesses of the paper are missing details on the proposed dataset (including how it was generated exactly), missing implementation details on the proposed training procedure (particularly the adaptive balancing of environments and learning rate), and a more thorough quantitative experimental validation of all claims, also in comparison to relevant recent related work.

Reviewers requested additional information during the disussion phase, but these requests were only met partially, with additional results promised after the end of the reviewing period.

Overall, the many weaknesses outweigh the strengths of this paper.
Particularly problematic is the missing quantitative experimental comparison against related work and the lack of quantitative results on controlled experiments (the qualitative results are not sufficient).

**Additional Comments On Reviewer Discussion:**

In the discussion, several points were raised, but only addressed partially:
- missing experimental comparison to relevant related work [Aug4] => source code of related work not public
- misleading claims on real-time performance [Aug4] => confirmed by authors
- missing details on geometry encoding [VGxx] => not provided
- unclear Figure 1 detailing architecture [Aug4] => not revised
- missing quantitative results on ablation experiments [Aug4] => not met sufficiently
- insufficient experimental results and analysis to validate claim on mode-collapse [Vgxx] => not met sufficiently

---

### Decision · Program_Chairs · 2025-01-22

Reject